# Diabetic Retinopathy and Skin Tissue Advanced Glycation End Products Are Biomarkers of Cardiovascular Events in Type 2 Diabetic Patients

**DOI:** 10.3390/jpm11121344

**Published:** 2021-12-10

**Authors:** Alejandra Planas, Olga Simó-Servat, Cristina Hernández, Ángel Ortiz-Zúñiga, Joan Ramón Marsal, José R. Herance, Ignacio Ferreira-González, Rafael Simó

**Affiliations:** 1Diabetes and Metabolism Research Unit, Vall d’Hebron Research Institute (VHIR), Vall d’Hebron University Hospital, Autonomous University of Barcelona, 08035 Barcelona, Spain; a.planas@vhebron.net (A.P.); olga.simo@vhir.org (O.S.-S.); cristina.hernandez@vhir.org (C.H.); a.ortiz@vhebron.net (Á.O.-Z.); 2CIBER de Diabetes y Enfermedades Metabólicas Asociadas (CIBERDEM), Spanish Institute of Health (ISCIII), 28029 Madrid, Spain; 3Cardiology Department, Vall d’Hebron University Hospital and Research Institute, Autonomous University of Barcelona, 08035 Barcelona, Spain; joseprmarsal@yahoo.es (J.R.M.); iferreir@vhebron.net (I.F.-G.); 4CIBER en Epidemiología y Salud Pública (CIBERESP), Spanish Institute of Health (ISCIII), 28029 Madrid, Spain; 5Medical Molecular Imaging Research Group, Vall d’Hebron Research Institute (VHIR), Nuclear Medicine Department, Vall d’Hebron University Hospital, CIBBIM-Nanomedicine, Autonomous University Barcelona, 08035 Barcelona, Spain; jherance@vhebron.net; 6CIBERBBN, Spanish Institute of Health (ISCIII), 28029 Madrid, Spain

**Keywords:** cardiovascular disease, advanced glycation end-products, diabetic retinopathy, cardiovascular disease biomarkers, type 2 diabetes, diabetic complications

## Abstract

Risk of cardiovascular events is not homogeneous in subjects with type 2 diabetes; therefore, its early identification remains a challenge to be met. The aim of this study is to evaluate whether the presence of diabetic retinopathy and accumulation of advanced glycation end-products in subcutaneous tissue can help identify patients at high risk of cardiovascular events. For this purpose, we conducted a prospective study (mean follow-up: 4.35 years) comprising 200 subjects with type 2 diabetes with no history of clinical cardiovascular disease and 60 non-diabetic controls matched by age and sex. The primary outcome was defined as the composite of myocardial infarction, coronary revascularization, stroke, lower limb amputation or cardiovascular death. The Cox proportional hazard multiple regression analysis was used to determine the independent predictors of cardiovascular events. The patients with type 2 diabetes had significantly more cardiovascular events than the non-diabetic subjects. Apart from the classic factors such as age, sex and coronary artery calcium score, we observed that the diabetic retinopathy and advanced glycation end-products in subcutaneous tissue were independent predictors of cardiovascular events. We conclude that the diabetic retinopathy and advanced glycation end-products in subcutaneous tissue could be useful biomarkers for selecting type 2 diabetic patients in whom the screening for cardiovascular disease should be prioritized, thereby creating more personalized and cost-effective medicine.

## 1. Introduction

Type 2 diabetes confers a substantial burden of macrovascular disease, with two-to four-fold increased risk of any cardiovascular event in comparison with non-diabetic patients [1]. Although type 2 diabetes is an independent risk factor for cardiovascular disease (CVD), not all patients with diabetes appear to be at equal risk. In fact, a high percentage of these patients will never experience vascular complications [2]. Therefore, early identification of diabetic patients at risk of developing CVD remains a challenge to be met [3,4]. 

It is well known that chronic hyperglycemia is related to cardiovascular (CV) complications of diabetes. However, the exaggerated risk for CVD in this population is not explained fully by conventional risk factors such obesity, hyperglycemia, dyslipidemia and hypertension, and, in fact, a substantial proportion of this risk remains unexplained [3,4]. Therefore, specific diabetes-related risk factors should be involved in the excess risk for CVD, and the tissue accumulation of advanced glycation end-products (AGEs) could be one of them. 

AGEs accumulate in the body during aging, and this process is accelerated by chronic hyperglycemia and oxidative stress [5], two conditions commonly present in type 2 diabetes. Therefore, the formation and accumulation of AGEs are accelerated by the diabetic milieu and contribute to vascular dysfunction and the accelerated development of atherosclerotic processes [6,7]. 

In recent years, a simple and non-invasive method for AGEs assessment through skin autofluorescence (SAF) has been validated [8] and related to the presence of micro- and macroangiopathy in individuals with type 2 diabetes [9]. In this regard, we previously reported that SAF was a good predictor of calcium score (CACs) > 400 AU, a reliable marker of coronary atherosclerosis [10].

Emerging data indicate that the presence of diabetic microvascular complications increases the risk of CVD [11,12]. In particular, diabetic retinopathy (DR) has been linked with an increase in risk for all-cause and cardiovascular mortality in patients with diabetes [12,13,14]. In this regard, it should be noted that diabetic-induced microvascular abnormalities that occur in the retina may also arise in other vascular beds, such as myocardial microcirculation [15,16]. However, DR is often missing as a risk factor in studies addressed to evaluate CVD. 

On this basis, the aim of this study is to evaluate whether the presence of diabetic retinopathy and accumulation of advanced glycation end-products (AGEs) in subcutaneous tissue can help to identify patients with type 2 diabetes at high risk of developing CV events and allow us to develop more personalized and cost-effective medicine.

## 2. Materials and Methods

### 2.1. Study Design and Subjects

This was a prospective case-control study comprising 200 subjects with type 2 diabetes and 60 non-diabetic controls matched by age and sex, all of them with no history of clinical CVD. The included subjects were enrolled in the PRECISED study (ClinicalTrial.gov accessed on 16 November 2021, NCT02248311).

All subjects enrolled met the following criteria: (a) type 2 diabetes diagnosed at least one year prior to the day of screening; (b) age from 50 to 79 years; (c) no history of vascular event; (d) no contraindication for the performance of CT scan or SAF assessment; and (e) no concomitant disease associated with a short life expectancy.

All included subjects were selected from the Outpatient Diabetic Clinic of Vall d’Hebron University Hospital and primary healthcare centers within its catchment area (North Barcelona). The recruitment period began on September 2014 and finished on June 2017. Of the 200 patients with type 2 diabetes, 13 withdrew consent, and the same occurred in 3 out 60 from the control group. Consequently, 187 subjects with type 2 diabetes and 57 non-diabetic controls were followed until December 2020.

The study was conducted according to the Declaration of Helsinki and was approved by the local ethics committee. All subjects provided written informed consent before study entry.

### 2.2. Data Collection and Laboratory Tests

Basal features of the subjects and classical cardiovascular risk factors (age, sex, ethnicity, current smoking, body mass index, systolic and diastolic blood pressure, clinical characteristic of diabetes disease and associated comorbidities) were collected during the first visit. In addition, a fasting venous blood sample was obtained from each recruited patient. Please see details in Appendix A.

### 2.3. Fundus Eye Examination

DR was evaluated by experienced ophthalmologists in mydriasis using slit-lamp biomicroscopy and retinography with the same camera (Topcon-DRI-OCTTRITON). The examiners classified DR according to the International Clinical Diabetic Retinopathy Disease Severity Scale [17]: (1) no apparent retinopathy, (2) mild non-proliferative retinopathy (NPDR), (3) moderate NPDR, (4) severe NPDR and (5) proliferative diabetic retinopathy (PDR).

### 2.4. Measurement of Skin Autofluorescence

SAF was measured using the AGE ReaderTM (DiagnOptics TechnologiesBV, Groningen, The Netherlands), a non-invasive desktop device. The AGE ReaderTM detects the characteristic fluorescence of some AGEs and was used to estimate the level of AGEs in the skin, and optical details of this non-invasive method have been described more extensively elsewhere [8] and are summarized in Appendix A. 

### 2.5. CT-CAC Scanning

First, the patient was prepared with beta blockers to decrease the heart rate, and nitroglycerin for vasodilatation if needed. Then, an ECG-synchronized prospective contrast-enhanced coronary CT was performed with SiemensBiograph mCT 64 s equipment (Siemens Healthcare GmbH, Erlangen, Germany). Automatic coronary vessel extraction of all coronary vessels with visual analysis of coronary stenosis was performed by researchers blind to the patient’s condition with “Syngo.Via” cardiac CT software, as described elsewhere [18]. The subjects were divided into two groups according their CACs: CAC < 400 AU and CAC > 400 AU. A value of CACs ≥ 400 AU was considered as “high coronary risk”.

### 2.6. Outcome

The primary outcome was the time to the first CV event. We defined a CV event as a composite of myocardial infarction, coronary revascularization, stroke, lower limb amputation or CV death.

### 2.7. Statistical Analysis 

Differences among groups were analyzed using Student’s *t*-test for quantitative variables with a normal distribution and Pearson’s chi-squared test for categorical variables. We calculated event-free survival according to the Kaplan–Meier method.

The Cox proportional hazard multiple regression analysis was used to determine independent predictors of CV events. Statistical analyses were performed with Stata statistical package 15. Significance was accepted at the level of *p* < 0.05 for all analyses.

## 3. Results

### 3.1. Basal Characteristics of the Sample 

The clinical characteristics and main laboratory findings of both groups (type 2 diabetes and controls), and the specific characteristics of subjects with type 2 diabetes, are shown in Table 1. We did not find any significant differences between groups regarding age, gender, ethnicity, smoking habit or family history of CVD. The specific characteristics of subjects with type 2 diabetes are shown in Appendix B (Table A1).

### 3.2. Follow-Up

187 subjects with type 2 diabetes and 57 non-diabetic controls were followed until December 2020. After a follow-up of 4.35 ± 1.43 years, a total of 24 CV events were registered, 23 CV events (12.3%) in type 2 diabetes group, and 1 (1.75%) in the non-diabetic control group. The Kaplan–Meier analysis shows vascular event-free survival regarding the groups (*p* = 0.031), (Figure 1).

In the type 2 diabetes cohort, we found an incidence rate of CV events of 28.2 per 1000 person years. The main basal clinical characteristics of patients with type 2 diabetes according to the presence of the primary outcome (the first vascular event) are shown in Table 2.

The multivariate Cox’s regression (Table 3), including the selected variables that were significant to the univariate analysis and well-known risk factors of CVD, showed that only age (HR 1.09, 95% CI 1.01–1.18, *p* = 0.024), gender (HR 0.35, 95% CI 0.15–0.83, *p* = 0.0174), the presence of DR (HR 2.58, 95% CI 1.14–5.85, *p* = 0.023), CACS > 400 AU (HR 4.16, 95% CI 1.14–10.26, *p* = 0.002) and a value of SAF on the 3rd tertile (HR 4.68, 95% CI 1.83–11.96, *p* = 0.001) were independently associated with the presence of a CV event.

## 4. Discussion

In the present study, we confirmed that the individuals with type 2 diabetes had significantly greater risk of having a CV event than the non-diabetic subjects. Furthermore, we provided evidence that DR and SAF (as a measure of tissue-AGE accumulation) were powerful predictors of CV events in the subjects with type 2 diabetes. 

Previously, we provided evidence that DR is an independent predictor of subclinical CVD [19] and that SAF is a good predictor of a CACs > 400 AU (a reliable marker of coronary atherosclerosis) [10]. The current study is important because we confirm that both DR and SAF are not only related to subclinical cardiovascular disease but are also useful in predicting CV events in type 2 diabetes population. 

According to our findings, previous reports have documented an increase in CV risk in patients with DR, mostly in those with advanced DR [20,21,22]. Although the underlying molecular mechanisms linking DR and cardiovascular disease are still a matter of debate, there are notable similarities in their pathophysiology. In this regard, recent evidence indicates that in individuals with type 2 diabetes, the vasa vasorum present evolutionary changes similar to those observed in the retina: an initial stage in which endothelial dysfunction and loss of capillaries predominate [16] and more advanced stages in which ischemia plays a key role, leading to angiogenesis and inflammation in response to the progressive enlargement of the necrotic core within the plaque [23]. This change in plaque phenotype results in a more inflamed and unstable plaque, favoring plaque rupture and a poor outcome of cardiovascular events. Thus, microcirculation represents a “common soil” between DR and CV events and would explain why DR is a good predictor of CV events. 

To the best of our knowledge, only two previous studies have examined the usefulness of SAF as a predictor of CVD in a prospective way [24,25]. Both support our data and conclude that SAF is strongly associated with the presence of CVD and cardiac mortality. Therefore, SAF could be a useful clinical tool to identify diabetic individuals who have a particularly high risk of developing CV events. It should be noted that our study is the first one conducted in subjects with type 2 diabetes without a history of clinical cardiovascular disease. This is important because it is precisely in this population, before the vascular damage becomes clinically apparent, for which we need useful biomarkers of CV risk.

In our study, we show that higher values of SAF were independently associated with the presence of macrovascular complications. Skin AGEs are accumulated mainly in collagen, which has low turnover and represents hyperglycemia over a longer time period than HbA1c. Therefore, SAF may reflect the impact of oxidative stress and the history of hyperglycemic episodes much better than the classical risk factors. In fact, SAF is considered a measure of metabolic memory in subjects with type 2 diabetes. 

In addition to DR and SAF, we found that other classical factors, such as age, male sex and CACs > 400 AU, were also related with the presence of a CV event. Age is an important determinant of CV risk, and it is known that the prevalence of inducible ischemia is significantly higher in type 2 diabetes patients over 65 years old [26]. Furthermore, it is well documented that the absolute risk of CV events is higher in men than women [27]. CACs is a well-recognized biomarker of myocardial ischemia and a good predictor of CV events [28]. In fact, guidelines recommend that the assessment of CACs could be considered in asymptomatic patients with diabetes mellitus who are over the age of 40 [3]. However, CACs assessment needs a CT scan examination, which can be inconvenient and rather expensive for routine practice in subjects with type 2 diabetes. For these reasons, retinal photographs, together SAF assessment, could replace CACs as biomarker of CV risk or, at least, be a useful tool to select those patients in whom CACs should be assessed. Noteworthy is the fact that Rima et al. recently demonstrated that deep learning and a retinal photograph-derived CAC score are comparable to CT scan-measured CAC in predicting a CV event [29]. 

The main limitation of our study is the low rate of CV events observed. However, it should be noted that there is a clear trend toward a decrease in events in diabetic subjects in the last 20 years, as reported by Rawshani et al. [30]. This is probably due to better management of chronic patients with diabetes, associated with better comprehensive control of the rest of the CV risk factors through the greater use of statins and antihypertensive drugs. 

## 5. Conclusions

In conclusion, this study reconfirms that patients with type 2 diabetes have significantly more CV events than non-diabetic subjects. In addition, DR and higher values of SAF are powerful predictors of CV events in subjects with type 2 diabetes and, therefore, might be included as meaningful variables in CV-risk stratification. Furthermore, DR and higher values of SAF could be useful biomarkers in selecting type 2 diabetic patients in whom the screening for cardiovascular disease should be prioritized, thereby generating more personalized and cost-effective medicine. 

## Figures and Tables

**Figure 1 jpm-11-01344-f001:**
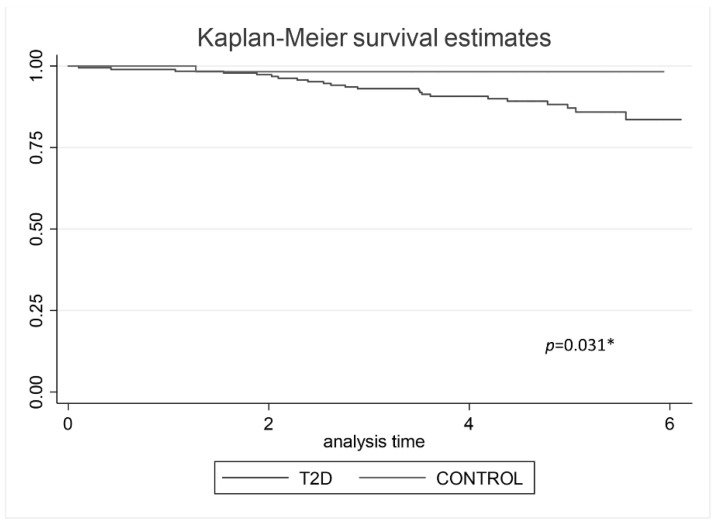
Kaplan–Meier analysis predicting vascular event-free survival regarding groups. * *p* value < 0.05 was statistical significative.

**Table 1 jpm-11-01344-t001:** Characteristics of subjects with type 2 diabetes and non-diabetic control subjects.

	Type 2 Diabetes (*n* = 187)	Control Group (*n* = 57)	*p*
Sex (woman) (*n*,%)	108 (57.75%)	37 (64.91%)	0.33
Ethnicity (Caucasian *n*,%)	179 (95.72%)	56 (98.25%)	0.65
Age (years)	65.63 ± 6.52	66.01 ± 6.63	0.85
BMI (kg/m^2^)	30.23 ± 4.89	26.83 ± 3.11	<0.001
Waist circumference (cm)	103.9 ± 13.53	91.2 ± 13.92	<0.001
Smoking			
No (*n*,%)	99(48.13%)	34 (59.65%)	0.59
Current Smoker (*n*, %)	25 (13.36%)	7 (12.3%)	
Ex-smoker (*n*, %)	62 (33.15%)	15 (26.32%)	
CV family history (*n*, %)	22 (11.76%)	8 (14.04%)	0.65
Hypertension (*n*, %)	135 (71.19%)	28 (49.12%)	0.001
Use of ACEi/ARB (*n*, %)	118 (63.1%)	18 (31.58%)	<0.001
Dyslipidemia (*n*, %)	149 (79.67%)	25 (43.86%)	<0.001
Use of statins (*n*,%)	133 (71.51%)	19 (31.67%)	<0.001
Use of ezetimibe (*n*,%)	10 (5.38%)	0	0.074
Total cholesterol (mmol/L)	4.78 ± 0.92	5.57 ± 0.91	<0.001
HDL cholesterol (mmol/L)	1.28 ± 0.32	1.28 ± 0.29	<0.001
LDL cholesterol(mmol/L)	2.72 ± 0.78	3.43 ± 0.81.14	<0.001
Triglycerides (mmol/L)	1.73 [0.50–5.67]	1.24 [0.46–5.27]	0.012
HbA1c (mmol/mol)	56.33 ± 9.01	42.02 ± 3	<0.001
HbA1c (%)	7.44 ± 1.19	5.55 ± 0.31	<0.001
Creatinine (mmol/L)	0.725 ± 0.021	0.067 ± 0. 0.017	0.075
GFR mL/min	81.76 ± 16.00	85.57 ± 10.88	0.09
AST (UI/L)	25.51 ± 15.71	23.48 ± 5.73	0.34
ALT (UI/L)	25.94 ± 16.88	21.12 ± 10.55	0.043
GGT (UI/L)	44.46 ± 71.82	31.04 ± 29.77	0.17
Skin AF (AU)	2.68 ± 0.65	2.41 ± 0.60	0.001
Log CACs	2.11 ± 0.81	1.59 ± 0.72	0.002
CCsA ≥ 400 AU (*n*, %)	41 (21.93%)	0	<0.001

**Table 2 jpm-11-01344-t002:** Clinical characteristics of patients with type 2 diabetes according to presence of primary outcome (first cardiovascular event).

	CV Event + (*n* = 23)	CV Event − (*n* = 164)	*p*
Follow up (*y*)	5.09 ± 1.20	5.21 ± 0.95	0.564
Sex (woman) (*n*, %)	8 (34.7%)	100 (60.9%)	0.017
Age (years)	68.61 ± 6.04	65.22 ± 6.49	0.019
BMI (kg/m^2^)	30.18 ± 4.19	30.23 ± 4.99	0.961
Diabetes duration (years)	17.69 ± 9.44	14.08 ± 9.34	0.084
Waist circumference (cm)	105.6 ± 11.89	103.69 ± 13.7	0.552
Smoking			0.943
No (*n*, %)	11 (47.8%)	88 (53.65%)	
Current smoker (*n*, %)	03 (13.04%)	22 (13.41%)	
Ex-smoker (*n*, %)	08 (34.37%)	55 (33.53%)	
Hypertension (*n*, %)	17 (73.9%)	118 (71.9%)	0.844
Dyslipidemia (*n*, %)	16 (69.76)	133 (81.1%)	0.198
Insulin treatment (*n*, %)	17 (73.9%)	91 (54.48%)	0.198
Fast plasma glucose (mmol/L)	7.99 ± 2.43	8.73 ± 2.79	0.232
HbA1c (mmol/mol)	58.45 ± 8.10	56.1 ± 9.08	0.234
HbA1c (%)	7.72 ± 1.07	7.41 ± 1.20	0.234
Total cholesterol (mmol/L)	4.69 ± 0.66	4.78 ± 0.95	0.682
HDL cholesterol (mmol/L)	1.33 ± 0.38	1.27 ± 0.30	0.399
LDL cholesterol (mmol/L)	2.73 ± 0.47	2.71 ± 0.82	0.906
Triglycerides (mmol/L)	1.39 [0.51–2.5]	1.53 [0.6–5.7]	0.046
Homocysteine (µmol/L)	12.5 [8.1–17.4]	11.3 [5.8–127]	0.765
Lipoprotein (a) (mg/dL)	7.21 [1–91.2]	8.45 [1–162.9]	0.745
GFR (mL/min)	86.5 ± 11.18	81.12 ± 16.46	0.285
Creatinine (mmol/L)	0.068 ± 0.01	0.0734 ± 0.02	0.278
Albumin/creatinine ratio			0.06
<3.39 mg/mmol (*n*, %)	9 (40.9%)	111 (68.5%)	
3.39–33.9 mg/mmol (*n*, %)	10 (47.6%)	44 (27.2%)	
>33.9 mg/mmol (*n*, %)	2(9.5%)	7 (11.3%)	
Log albumin/creatinine ratio	1.50 ± 0.70	1.25 ± 0.61	0.085
Diabetic Retinopathy (*n*,%)	11 (47.82%)	40 (24.40%)	0.018
Diabetic Neuropathy (*n*,%)	3 (13.04%)	32 (19.451)	0.450
CACS > 400 AU (*n*, %)	10 (52.63%)	31 (19.562)	0.001
Log CACs (AU)	2.55 ± 0.84	2.05 ± 0.78.7	0.013
AGEs 3rd Tertil (AU)	12 (63.15%)	39 (26.71%)	0.001
AAS (*n*,%)	6 (27.27%)	54 (32,92%)	0.594
Statines (*n*,%)	14 (63.63%)	119 (72567%)	0.384

**Table 3 jpm-11-01344-t003:** Results of multivariate Cox’s regression for predicting vascular event.

	HR	CI 95%	*p*
Sex (female)	0.35	0.15–0.83	0.017
Age (years)	1.09	1.01–1.18	0.024
BMI (kg/m^2^)	0.99	0.91–1.08	0.820
Diabetes duration (years)	1.04	0.99–1.08	0.093
Waist (cm)	1.01	0.98–1.04	0.526
Hypertension (yes)	1.13	0.45–2.88	0.792
Dyslipedemia (yes)	0.59	0.24–1.44	0.244
Insulin treatment (yes)	2.11	0.83–5.36	0.116
HbA1c (mmol/mol)	1.20	0.88–1.66	0.255
GFR (mL/min)	1.02	0.99–1.05	0.170
Creatinine (mg/dL)	0.33	0.04–2.44	0.275
Diabetic Retinopathy (yes)	2.58	1.14–5.85	0.023
CACS > 400 AU (yes)	4.16	1.69–10.26	0.002
AGEs 3rd Tertil (yes)	4.68	1.83–11.96	0.001

## Data Availability

The data presented in this study are available on request from the corresponding author.

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
