# Peer review of "Diabetic Retinopathy and Skin Tissue Advanced Glycation End Products Are Biomarkers of Cardiovascular Events in Type 2 Diabetic Patients"

_jpm, 2021, doi:10.3390/jpm11121344_

Round 1

Reviewer 1 Report

The association of diabetes with cardiovascular disease, especially atherosclerosis, is well known. The occurrence of diabetic complications - diabetic micro- and macroangiopathy, including retinopathy, is associated with vascular pathology. New markers of vascular and cardiovascular disease, including AGEs, have been studied for several years. In further studies, attention should be paid to the high selectivity and sensitivity of testing in specific vascular diseases, including atherosclerosis so closely related to diabetes and its complications.

Author Response

Many thanks for your comments. Although AGEs has been involved in cardiovascular disease, there is little information regarding the usefulness of AGEs assessment through skin autofluorescence (SAF) to predict cardiovascular events. We would like to emphasize that in present study we demonstrated that SAF is a powerful predictor of CV events in subjects with type 2 diabetes and, therefore, could be included as meaningful variables in CV risk stratification.

Reviewer 2 Report

This article might have clinical implications. It is well written.

Page 8 line 249:

The authors have said: In conclusion, this study confirms that patients with type 2 diabetes have significantly more CV events than non-diabetic subjects.

I would say: ….reconfirms, not confirms because this is very well known.

Page 8 line 252:  Instead of ``….could be included as meaningful variables in CV risk stratification.`` I would say: ``…might be include….``. I consider that is too early to say that.  

References are not many; some of these are very old. 

Page 2: line 76 and 83 has the same acronym (DR).

Author Response

This article might have clinical implications. It is well written.

- Thank you very much for the revision process and you constructive criticism on our paper.

Page 8 line 249:

The authors have said: In conclusion, this study confirms that patients with type 2 diabetes have significantly more CV events than non-diabetic subjects.

I would say: ….reconfirms, not confirms because this is very well known.

- We have change the wording of this sentence as suggested.

Page 8 line 252:  Instead of ``….could be included as meaningful variables in CV risk stratification.`` I would say: ``…might be include….``. I consider that is too early to say that.  

- The sentence has been changed as suggested.

References are not many; some of these are very old. 

- I added 2 more references. 

Page 2: line 76 and 83 has the same acronym (DR).

  • Done!

I submit my last version of my manuscrpt.

thank you for your comments.

Alejandra
